# Research on the Development Law of Pre-Mining Microseisms and Risk Assessment of Floor Water Inrush: A Case Study of the Wutongzhuang Coal Mine in China

Lei Huang [1] , Yanchun Xu [1], Shiqi Liu [1,2,*], Qiukai Gai [3,*], Wei Miao [1], Yubao Li [4] and Lisong Zhao [4]

1   School of Energy and Mining, China University of Mining and Technology (Beijing), Beijing 100083, China
2   Jizhong Energy Group Co., Ltd., Xingtai 054000, China
3   State Key Laboratory for Geomechanics & Deep Underground Engineering, China University of Mining and Technology (Beijing), Beijing 100083, China
4   Hebei Coal Research Institute Co., Ltd., Xingtai 054000, China
*   Correspondence: 201845@cumtb.edu.cn (S.L.); geqiukai@163.com (Q.G.)

**Abstract:** Coal-mining areas are widely distributed in Northern China, but are under threat from confined water in the mining operation, resulting in a series of floor water- inrush hazards. Therefore, it is significant to effectively evaluate the floor water inrush to ensure safe and efficient coal mining. The 182602 working face of the Wutongzhuang coal mine served as the background for our research. The concept of "pre-mining microseisms" was proposed, and based on this, microseismic monitoring equipment was arranged on site. The correlation between microseismic events and the water abundance of an aquifer was analyzed, and a floor water inrush evaluation method was constructed based on the three elements of an aquifer and pre-mining microseisms. The main results are as follows: first, the microseismic events were excited by artificial disturbances before the mining of the working face including slurry diffusion and neighboring mining, which had the characteristics of sporadicity, clustering, and periodicity. Second, the regional distribution of water abundance was determined by taking the water inflow, water pressure, and grouting volume as the outward performance characteristics of water abundance in the Shanvuqing aquifer. Furthermore, the correlation coefficient between the pre-mining microseisms and the three elements of the aquifer (water inflow, water pressure, and grouting volume) was larger than 0.7. On this basis, an evaluation method associated with the water inrush risk along the strike of the working face was established based on pre-mining microseisms, dividing the working face into dangerous zones, suspected dangerous zones, and safe zones. Furthermore, pre-mining microseisms, water abundance, and structures were introduced as risk-evaluation indices, and the complete weight was calculated using an analytic hierarchy process and entropy-weight technique, before a vulnerability index model of floor water inrush was built. Finally, targeted treatment procedures were efficiently implemented to ensure the safe mining of working face 182602 due to the successful prediction of potential water risk zones. The research results provide scientific and technological support for pre-mining microseisms combined with water abundance as a technical method to prevent floor water inrush.

**Keywords:** pre-mining microseisms; floor water inrush; water abundance; forecast

## 1. Introduction

Coal is the main basic energy source in China and plays a leading role in the primary energy structure. China's coal consumption in 2020 was 4.04 billion tons, and this is expected to reach as high as approximately 3.6 billion by 2030 [1,2]. At present, with the large-scale mining of coal resources, the current mining conditions have changed from easy to difficult, especially due to the prominent problem of floor water inrush. It is estimated that more than 55% of coal mines in China are limited by floor-confined water [3]. In recent years, most scholars have focused their attention on the study of the

mechanism and evaluation method of floor water inrush including "in situ fissures" and the "original destruction" theory [4], the "five figures plus double coefficients" method [5], the key layer "KS" theory of floor water inrush [6], the mechanism of delayed water inrush from the floor under the combined action of confined water [7], the Fisher's discriminant model [8], and the height prediction of the water-flowing fracture zone [9]. In addition, many numerical simulation methods for the coupling of engineering-scale mining-water invasions have been obtained to reveal the progressive failure mechanisms of floor rock or coal pillars under mining-water immersion [10–12]. However, there are still deficiencies in the equipment for monitoring floor water inrush, and the development of the technology lags seriously behind the field's urgent needs. Conventional monitoring technology has the disadvantages of poor monitoring timeliness, a small monitoring range, poor continuity, and a poor monitoring effect. As a result, an effective technology is urgently needed to compensate for the deficiencies of water inrush monitoring and early warning in coal mines [13].

A microseismic monitoring system was adopted to monitor the stability of engineering rock prior to deformation and damage. This technology was previously used primarily to monitor high-energy dynamic disasters in coal mines such as rock bursts and rock weighting [14]. Nearly half of the coal-mine disasters in China have been found to occur in clusters or be accompanied by earthquakes nearby, in which all the disaster types are involved [15]. As a matter of fact, the microseismic monitoring system could also detect the tiny vibration signals generated by rock failure during the formation of water-conducting channels in coal mines. It was also used to evaluate the risk of water inrush by introducing specific parameters such as the time and location of the water-conducting channel. Microseismic monitoring was first undertaken in a deep mine in China where water inrush was a significant issue for the progressive failure of the geological structures under investigation. With the results of the microseismic events and 3D illustration techniques, the activities of the geological structure and the fracture depth of the roof and floor were obtained [16]. Based on in situ experiments and numerical simulations, a new academic idea was proposed: "The rock microseismicity induced by mining processes and water pressure disturbance is in essence the index of groundwater inrush". Groundwater inrush models have been calibrated and could be used to clarify the precursory characteristics and to locate the inrush pathway [17]. The microseismic monitoring technique was applied to the floor-mining fault, and a floor-fault water inrush prediction model was proposed based on the catastrophe analysis of microseismic signals [18]. Based on these academic results, the key layer of the roof was divided into two boundaries; taking the Dongjiahe coal mine in China as an example, the space–time and energy distribution of microseismic events were analyzed and this model of a coal mine roof was studied along the vertical direction [19]. This microseismic monitoring technique has been used successfully to evaluate the risk of water inrush in several mines such as the Tashan coal mine (Datong, China), Jiulishan coal mine (Jiaozuo, China), Fucun coal mine (Zaozhuang, China), and Halagu coal mine (Erdos China).

The findings of the aforementioned research have aided in the development and use of microseismic monitoring systems in areas such as groundwater inrush early warning. Nevertheless, previous research on microseismic characteristics and their engineering applications in coal mines has primarily focused on the distribution of microseismic events on the roof and floor during the mining process. Less is known about the characteristics of microseismic events before mining, which can be called 'pre-mining microseisms.' In this paper, taking the 182602 working face of the Wutongzhuang coal mine in China as an example, the authors statistically analyzed the distribution patterns of the floor pre-mining microseismic events. The hydrogeological characteristics on the working face caused by pre-mining microseismic could be determined, then the spatial location with high water abundance was judged. The microseismic activity information in the working face and surrounding area was interpreted in detail, and the characteristics of pre-mining microseismic were summarized. The pre-mining microseismic data were turned into valuable hydrogeological characteristic response data, which was normally deemed useless

in the grouting process. The research results further provide the basis and reference for microseismic data analysis and water inrush risk evaluation under similar conditions.

## 2. The Study Area and Its Engineering Background

### 2.1. Geological Conditions of 182602 Working Face

As shown in Figure 1, the Wutongzhuang coal mine, belonging to the mining area of Handan, is situated southeast of Handan City, Hebei Province, China, which is approximately 35.0 km from the center of Handan City. The 182602 working face with an average depth of 850 m, is located in coal seam 2 of the sixth mining area in the Wutongzhuang coal mine. The structure of the coal seam is simple, with an average thickness of 3.20 m, a hardness of 0.6, and an average dip angle of 14°. An inclined comprehensive long-wall stratified mechanized mining method was applied at the 182602 working face, that is, the width of the working face is 256 m, the length of the working face is 834 m, the average mining height is 3.5 m, and the average daily mining speed is 2.9 m/d. Normal faults are relatively developed inside the working face, and six faults and one syncline in total were found in the haulage gateway and air-return gateway. Through exploration, the fault does not conduct water, but the syncline conducts water. The layout of the 182602 working face is shown in Figure 2. In addition, the 182602 working face is adjacent to the 182101 working face with similar geological conditions.

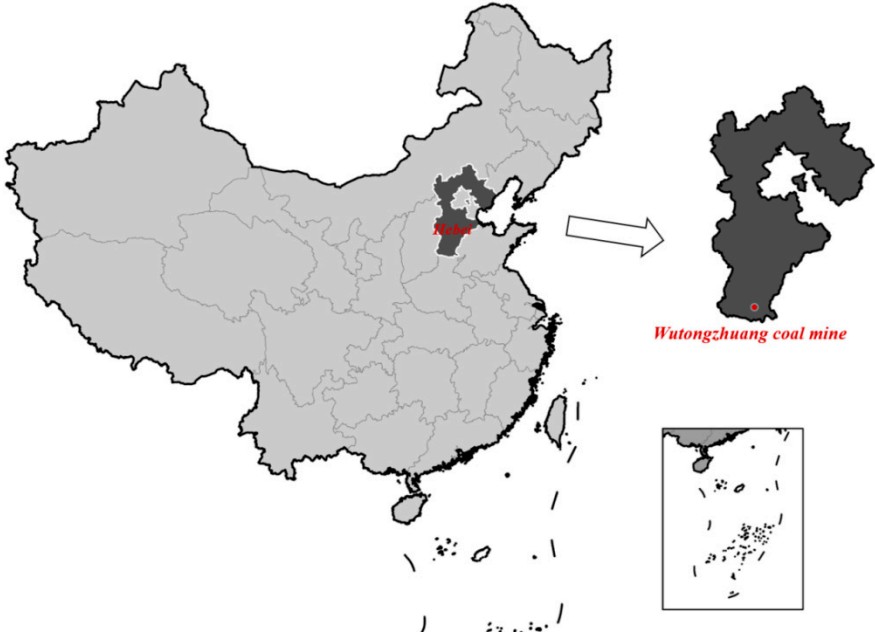

**Figure 1.** The location of the Wutongzhuang coal mine.

The immediate roof is sandy shale, with an average thickness of 3.21 m, whereas the main roof is fine-grained sandstone, with an average thickness of 6.13 m. The seam's immediate floor is sandy shale, with an average thickness of 3.54 m, whereas the seam's basic floor is fine sandstone, with an average thickness of 7.10 m. The lithological description of the coal seam roof and floor is listed in Figure 3.

The direct water-filled aquifer of the floor is the Yeqing limestone aquifer with an average thickness of 2.5 m and a hydraulic pressure of 0.35~7.2 MPa, which is approximately 38 m away from the coal seam. Another direct water-filled aquifer of the floor is the Shanvuqing limestone aquifer, with an average thickness of 5.5 m and a hydraulic pressure of 1.3~8.8 MPa, which is approximately 58.0 m away from the coal seam. The water bursting coefficient of the Shanvuqing aquifer is 0.07 MPa/m, which is greater than the critical value of 0.06 MPa/m. The Ordovician limestone aquifer has a wide distribution area, large thickness, and high water pressure, which is a major threat to mine safety production. The geological description of the aquifers is listed in Table 1.

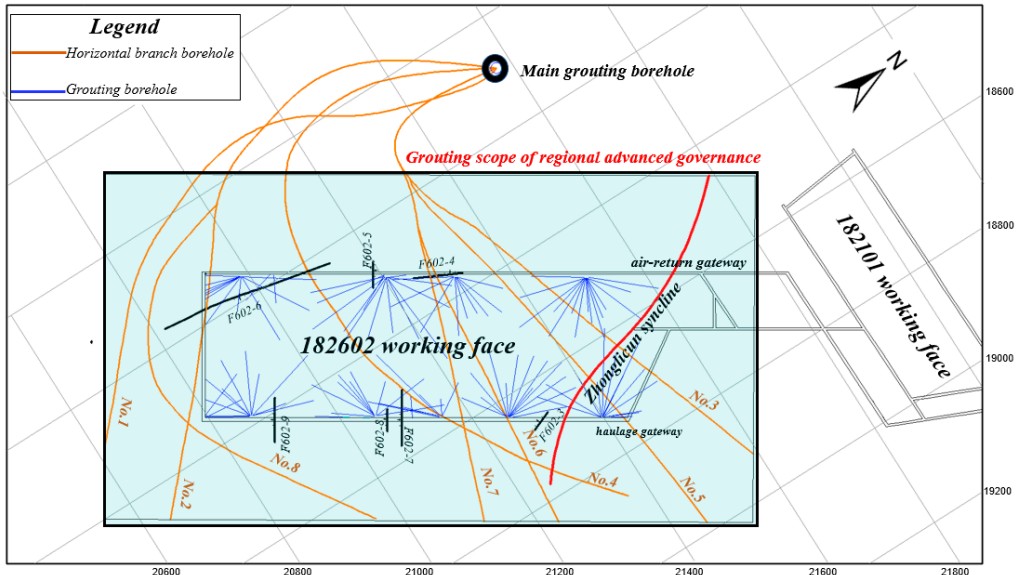

**Figure 2.** The layout of the 182602 working face in the Wutongzhuang coal mine.

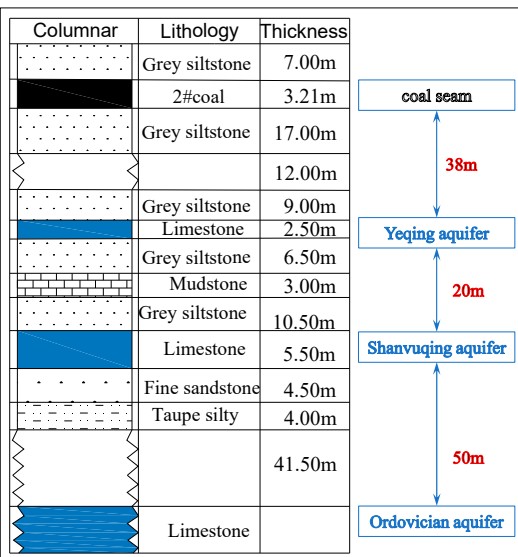

**Figure 3.** The lithological description of the coal seam roof and floor.

**Table 1.** The geological description of the aquifers.

| Aquifer | Average Thickness (m) | Distance from Coal Seam(m) | Water Pressure (MPa) | Permeability Coefficient (m/d) | Specific Capacity (L/s·m) | Water Quality Type | Mineralization (g/L) |
|---|---|---|---|---|---|---|---|
| Yeqing | 2.5 | 38 | 0.35~7.2 | 0.04 | 0.02 | Cl.SO4-Na. Ca | 4.3 |
| Shanvuqing | 5.5 | 58 | 1.3~8.8 | 2.35 | 0.078 | Cl.SO4-Na. Ca | 5.5 |
| Ordovician | - | 108 | 5.3~9.6 | 5.89 | 1.258 | Cl.SO4-Ca. Na | >5.24 |

## 2.2. Overview of the Regional Advanced Governance Technology on the 182602 Working Face

In Chinese mines, two different types of water prevention and control countermeasures are typically employed. One is the method of hydrophobic depressurization, which helps to lessen the risk of floor water inrush. However, when the water abundance of the aquifer is strong and the recharge is large, it will run into issues with reducing the hydraulic head difficulty, the high water pressure or drainage, and high cost. Therefore, few mines are currently using this method. Another method is the regional advanced governance

technology, which can increase the thickness of the aquifuge, and enhance the water insulation and rock strength of the floor rock mass [20].

The regional advanced governance technology was used to inject slurry into the grouting hole, which can be solidified according to the designed concentration. It would press and convey the slurry to the prescriptive rock and soil, and then fill in the soil cracks or pores, forming a similar curtain of the concrete cut-off wall [21,22].

The Wutongzhuang coal mine is a typical coalfield with a high risk of water leakage in Northeast China, which has adapted the regional advanced governance technology to pre-dispose the water leakage in the Shanvuqing and Ordovician limestone. As shown in Figure 2, the key to achieving regional advanced governance technology is ground multi-branch horizontal directional grouting and floor grouting reinforcement.

The ground multi-branch horizontal directional grouting project was carried out on the 182602 working face. One main grouting borehole and eight horizontal branch grouting boreholes were constructed on the ground face, and the grouting horizon of the horizontal branch boreholes was the Shanvuqing aquifer. The information on the ground grouting boreholes is shown in Table 2. Furthermore, in the floor grouting reinforcement project, a total of 174 boreholes were constructed to expose the Shanvuqing limestone aquifer, and the cumulative grouting volume was 998.77 tons.

**Table 2.** The information on ground horizontal branch boreholes.

| Borehole Number | Horizontal Length (m) | Grouting Time | Grouting Volume (t) | Circulation Loss (t) |
|---|---|---|---|---|
| No. 1 | 739 | December 2017 | 159 | - |
| No. 2 | 811 | January 2018 | 295 | - |
| No. 3 | 760 | March 2018 | 16 | - |
| No. 4 | 652.5 | June 2018 | 376 | 40 |
| No. 5 | 866 | July 2018 | 437 | 28 |
| No. 6 | 717 | August 2018 | 82 | 13 |
| No. 7 | 700 | September 2018 | 124 | - |
| No. 8 | 764 | September 2018 | 52 | - |

*2.3. Overview of the Microseismic Monitoring System on the 182602 Working Face*

The KJ1073 microseismic monitoring system, which was created independently by the Key Laboratory of Mining Microseismic in Hebei Province, China, may be utilized for daily monitoring on the 182602 working face. The system is made up of both ground and underground components. The ground equipment consists mostly of the 'data processing computer', 'mainframe for monitoring', and underground equipment.

To create an 'encircled' geophone array structure and a high-precision microseismic monitoring system, we attempted to install geophones at various locations and optimizing the placement algorithm, ultimately determining the following based on the installation and debugging results. As shown in Figure 4, five single-axis geophones and one triaxial geophone were placed on the haulage gateway of the 182602 working face, and six single-axis geophones and one triaxial geophone were installed in the air-return gateway, with a distance of 130 m between each geophone.

The 182602 working face was mined on 7 November 2018. The microseismic monitoring system was set up and monitored on 30 May 2018. The total time of the pre-mining microseismic monitor was five months. Microseismic events occurred between the No. 2 coal seam floor and the Ordovician limestone karst aquifer before the mining of the 182602 working face, which were characterized by "large frequency, large scope, many aggregation zones, and dispersion" due to the regional advanced governance project and the mining disturbance of the adjacent 182101 working face. These are referred to as "pre-mining microseisms" as a whole.

The pre-mining microseismic monitoring process was divided into two periods based on the stoppage time of the adjacent 182101 working face (29 July 2018). The first period

was from June to July 2018. The second period involved August, September, and October 2018.

**Figure 4.** The microseismic monitoring system arrangement on the 182602 working face.

## 3. Dynamic Development Pattern and Distribution Characteristics of Pre-Mining Microseisms

### 3.1. Plane Distribution Characteristic of Microseismic Events within the Range of Shanvuqing Aquifer

The kernel density estimation was used to analyze the plane distribution characteristic of the microseismic events. The kernel density cloud chart of microseismic events within the range of the Shanvuqing aquifer during the first period is shown in Figure 5, and the kernel density cloud chart of the microseismic events during the second period is shown in Figure 6. Different degrees of blue in the figures represent the different density values of the microseismic events. Referring to the results of other studies and the empirical analysis, the zone with a frequency kernel density greater than $0.00006/m^2$ was the fracture developing zone or potential zone of the water-conducting channel under the action of stress disturbance.

During the first period, the kernel density of microseismic events was distributed in a V-shaped continuous two-center type with the Zhonglicun syncline as its axis. The high-frequency kernel density center 'A' was located in the zone near the main grouting borehole, and the other kernel density center 'B' was located in the zone near the 182101 working face. The maximum kernel density value during June and July was $0.000339/m^2$, and the kernel density zone extended in a belt from the center to the 182602 working face. The maximum kernel density value within the 182602 working face was $0.00024/m^2$, indicating that the fractures within the working face had not been effectively blocked, especially in the zone near the Zhonglicun syncline.

During the second period, the kernel density distribution of microseismic events showed a phase change. The kernel density zone during August 2018 showed a bicentric discontinuous distribution in a northwest–southeast direction. When compared to August 2018, the kernel density zone during September 2018 showed a one-center discontinuous distribution in the northwest–southeast direction, with a 19.2% increase in the center 'A' kernel density value and a 41.3% decrease in the center 'B' kernel density value. When compared to September 2018, the kernel density area during October 2018 showed a non-center discontinuous distribution in the northwest–southeast direction, with a 45.1%

decrease in the center 'A' kernel density value and a 41% decrease in the center 'B' kernel density value. There was no microseismic event response within the 182602 working face during the second period, indicating that the fracture within the 182602 working face had been effectively sealed. The kernel density value of center 'A' increased or decreased due to the influence of the grouting time and grouting volume of the horizontal branch grouting boreholes, and the kernel density value of center 'B' gradually decreased due to the stoppage of the adjacent 182101 working face.

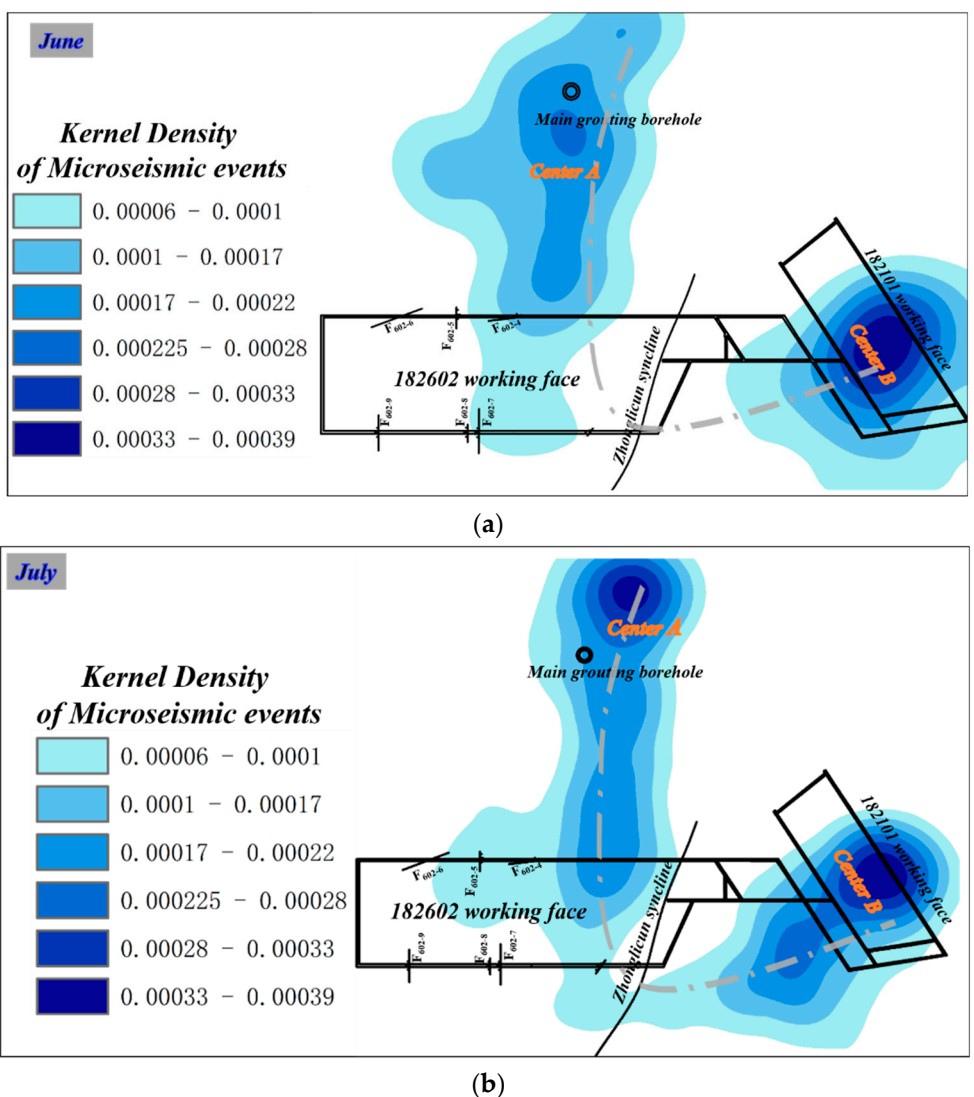

**Figure 5.** Kernel density distribution characteristic of microseismic events in the first period. (**a**) Kernel density value during June. (**b**) Kernel density value during July.

The aggregation of microseismic events in center 'A' reflected the expansion process of fractures caused by the high-pressure slurry, and the aggregation of microseismic events in center 'B' reflected the expansion process of fractures caused by mining stress disturbance in the adjacent 182101 working face. During the monitoring process, the kernel density value of center 'B' showed a decreasing trend. It was determined that the decrease in the kernel density values of the microseismic event was related to the progressive decrease in the residual mining stress following the stoppage of the 182101 working face, resulting in the gradual closure of deep water-conducting fractures. During the monitoring process, the kernel density value of center 'A' increased first and then decreased. The high-pressure slurry was determined by expanding the hydraulic fractures and cavities, closing them, releasing significant strain energy, and causing microseismic events. However, the fractures

were filled and strengthened with the progress of grouting, some vertical and horizontal water-conducting channels were blocked, and the groundwater flow field was rebalanced, leading to a greater decrease in the nuclear density of microseismic events, especially in October 2018.

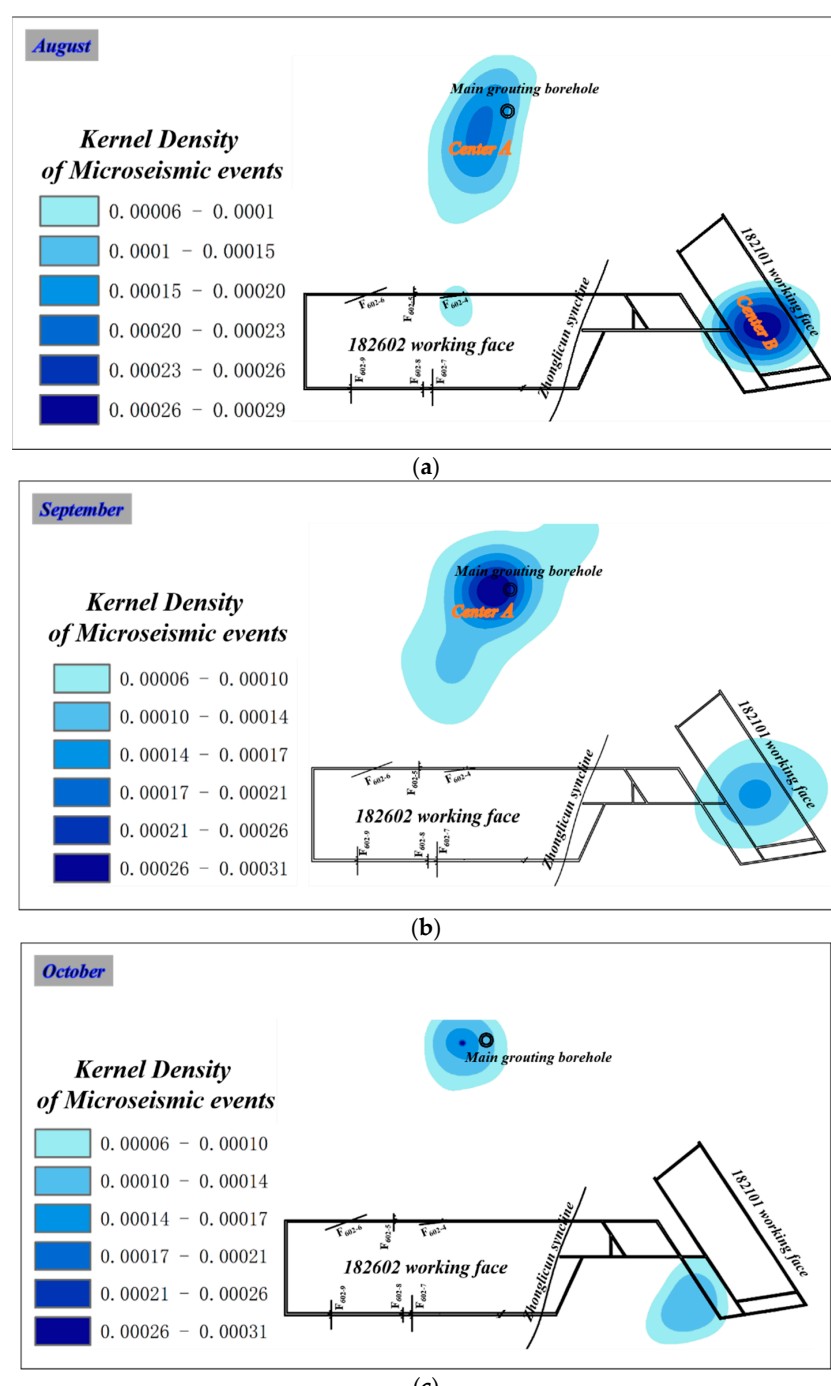

**Figure 6.** Kernel density distribution characteristic of microseismic events in the second period. (**a**) Kernel density value during August. (**b**) Kernel density value during September. (**c**) Kernel density value during October.

Therefore, comparing the changes in the distribution of the microseismic kernel density values between the first and second periods, it can be seen that pre-mining microseisms would be excited only after the equilibrium of the rock–fluid system was disrupted under artificially disturbed conditions (high-pressure slurry diffusion, neighboring mining stress)

in the working face's pre-mining phase. When the influence of external disturbances weakens, the aggregation degree of microseismic events would weaken until it disappears.

### 3.2. Spatial Distribution Characteristic of Microseismic Events within the Range of Shanvuqing Aquifer

The microseismic events within the Shanvuqing aquifer were spatially located and the spatial distribution chart of the microseismic events was drawn. The red sphere represented microseismic events and the gray surface represented a coal seam. In addition, the XZ-axis and YZ-axis directions of the space were projected, respectively. Taking June 2018 as an example, the spatial distribution of microseismic events in the Shanvuqing aquifer is shown in Figure 7.

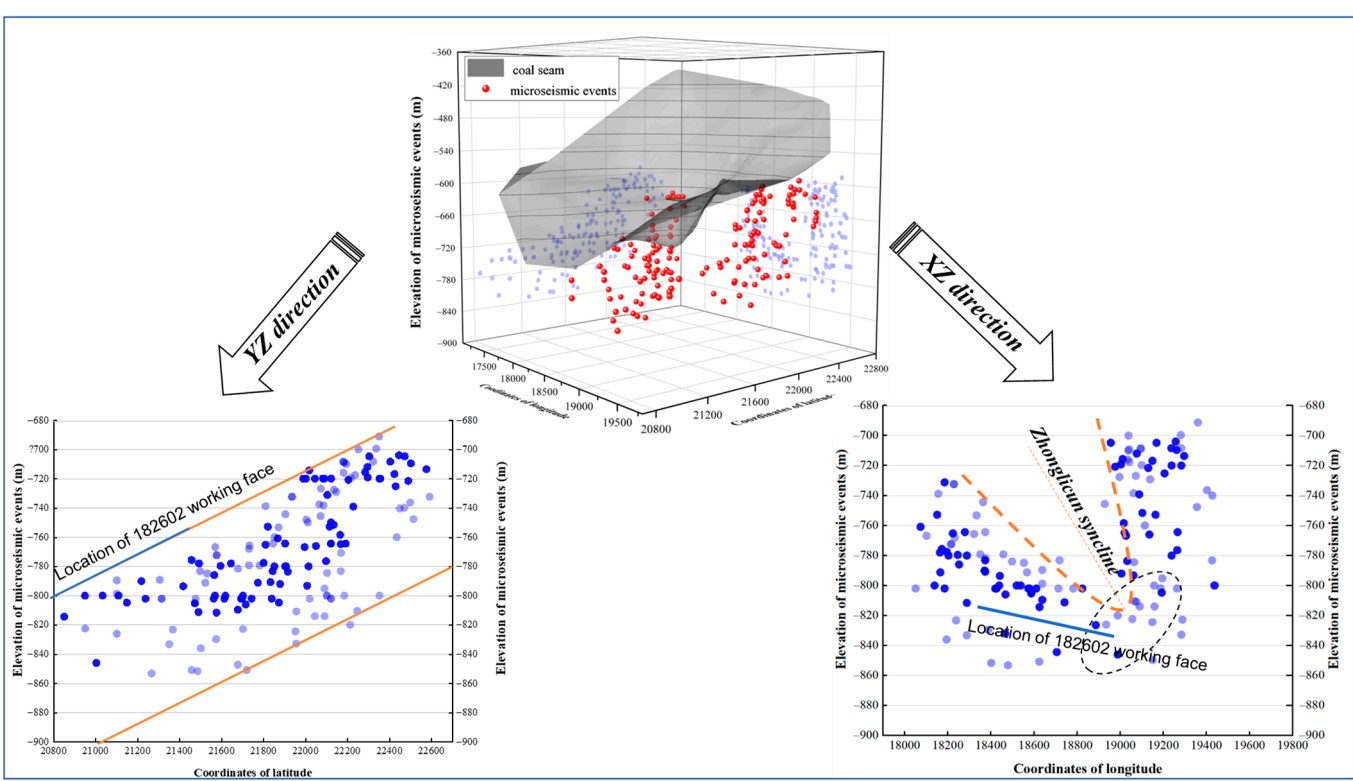

**Figure 7.** The spatial distribution of microseismic events during June 2018.

The coal seam elevation in the monitoring area ranged from −950 m to −360 m, and there were many microseismic events within the Shanvuqing aquifer and spatially distributed continuously. The microseismic events that were projected took the Zhonglicun syncline as the axis in the XZ-axis direction, showing a "V" continuous distribution. There were massive microseismic events at the endpoint. These were distributed in a belt along the two wings of the Zhonglicun syncline, with a left wing width of 338.8 m and a right wing width of 348.3 m. The microseismic event projection was distributed in an inclined strip along the axial plane of the Zhonglicun syncline in the YZ-axis direction, with a width of 313.2 m.

The spatial distribution characteristics of the pre-mining microseisms during July, August, September, and October 2018 were similar. The microseismic event projection was distributed in an inclined strip along the axial plane of the Zhonglicun syncline in the YZ-axis direction. The spatial distribution of microseismic events showed that the axial part and the two wings of the Zhonglicun syncline were more fractured, and more strain energy had accumulated inside due to the influence of high-level tectonic stress. Under the action of tectonic stress, high ground stress, and high water pressure, strain energy was released, which made fractures and cavities with water storage space develop and expand,

leading to the destruction of the groundwater flow field equilibrium, thus causing rock damage and exciting microseismic events.

As a result, the higher the possibility of the existence of a potential water-conducting channel in zones with a high degree of fissure development (the two wings of Zhonglicun syncline), the rock system's equilibrium was more likely to be disrupted, and the pre-mining microseismic events often aggregated in these types of zones.

### 3.3. Energy Evolution Characteristics of Microseismic Events within the Range of Shanvuqing Aquifer

According to the previous dynamic phenomena, when the energy release of a single microseismic event reaches more than $10^4$ J, the rock mass may be macroscopically damaged [23]. A large microseism was defined as when the energy release of a single microseismic event was greater than $10^4$ J, and a destructive micro-seismic as when the energy release of a single microseismic event was greater than $10^5$ J, based on the pertinent literature and the microseismic energy release in the Wutongzhuang coal mine. The destructive microseisms reflected the instantaneous rupture of the coal and rock mass, accompanied by the instantaneous release of accumulated elastic energy.

The construction of the No. 4 grouting borehole was on 9 June 2018, and the construction of the No. 5 grouting borehole started on 26 July 2018. To analyze the effect of grouting on the excitation of microseismic events, we counted the representative microseismic monitoring data from 10 June to 20 August 2018 before mining and quantified the energy of the microseismic events. A total of 900 microseismic events were collected. As shown in Figure 8, the grouting impact period was divided into three phases regarding the energy variation characteristics of the microseismic events.

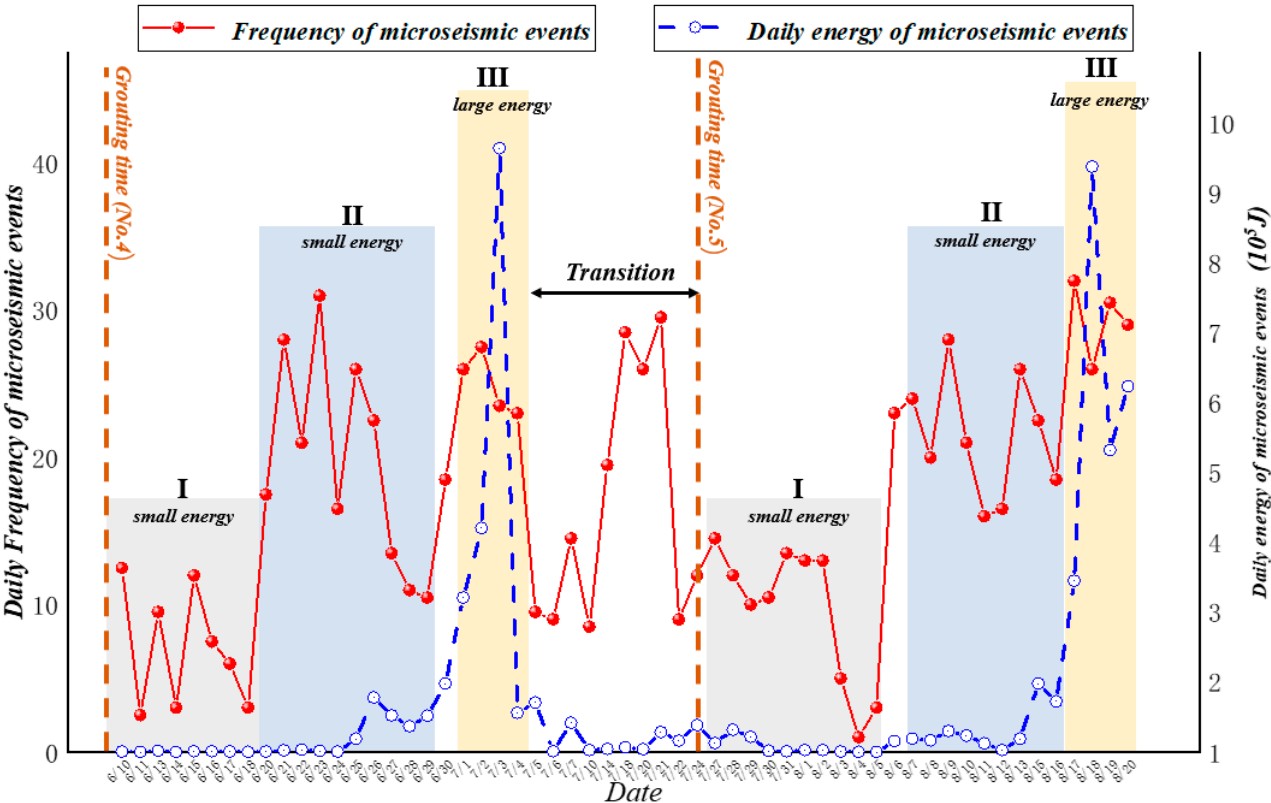

**Figure 8.** The energy evolution characteristics of the microseismic events.

The first phase lasted nine days, from 10 June to 18 June, and the microseismic events were characterized by 'low frequency and small energy'. The daily frequency of microseismic events was less than 15, the daily energy was less than $10^5$ J, and the energy of a single microseismic event was less than $10^3$ J. The second phase lasted ten days, from

20 June to 29 June, and the microseismic events were characterized by 'high frequency and small energy'. The daily frequency of microseismic events was not less than 15, but the daily energy was not more than $5 \times 10^5$ J, and 95.8% of the energy of microseismic events was less than $10^3$ J. The third phase lasted four days, from 1 July to 4 July, and the microseismic events were characterized by 'high frequency and large energy'. The duration of this phase was short. The daily frequency of microseismic events was generally greater than 20, and the daily energy exceeded $5 \times 10^5$ J. Then, from 5 July to 25 July was the transition, which usually lasted for a long time, and there were no grouting boreholes during this period. The daily frequency of microseismic events varied greatly, but the daily energy was small, with the maximum not exceeding $5 \times 10^5$ J. The energy of 87.4% of microseismic events was less than $10^3$ J. Until the construction of the No. 5 grouting borehole (26 July 2018), the distribution of microseismic events repeated the above phases in turn, and the frequency and energy changed periodically.

The characteristics of the pre-mining microseisms varied with the progression of grouting, which was obviously different from the large-scale microseisms induced by periodic weighting after mining of the working face. The pre-mining microseisms showed a phased change, and the inducement was mainly a small-scale microfracture. In the first phase, the slurry was naturally filled into large fractures or rock defects, the grouting time was long, there was no need for grouting pump pressure, and the stress concentration caused by filling grouting was small, resulting in the microseismic events characterized by 'low frequency and small energy'. In the second phase, the grouting time was long, the high-pressure slurry diffused radially, destabilizing the groundwater flow field, and the penetrative grouting induced stress concentration and strain energy release, resulting in microseismic events characterized by 'high frequency and small energy'. In the third phase, the grouting time was short, the high-pressure slurry caused further expansion of the fracture opening, fractures of the grouting injured the rock mass, and a substantial amount of strain energy was released, resulting in the microseismic events characterized by 'high frequency and large energy'. When the energy was concentrated and released, it reentered a new round of energy accumulation and release process. The microseismic events were excited again until the next grouting, and their characteristics changed in phase.

### 3.4. Dynamic Development and Distribution Characteristics of Pre-Mining Microseisms

Combined with the analysis of plane distribution, spatial distribution, energy distribution, and the disturbance factors of microseismic events before mining, we found the following three interesting performance characteristics of the pre-mining microseisms.

One of the characteristics of pre-mining microseisms was sporadicity. There were periods of high microseismic activity followed by periods of relative calm. The sporadicity of microseismic activity reflected the regional stress or local rock mass stress anomaly. For example, the frequent microseismic events at center A in June and July 2018 were closely related to the branch hole grouting activity at this location. However, the substantial decrease in microseismic events at center A during October 2018 was associated with the decrease in the strain energy release triggered by slurry diffusion near the end of grouting. Therefore, the sporadic analysis of pre-mining microseisms can be used for real-time monitoring and evaluation of the grouting engineering effect, and guide the next grouting work.

One of the characteristics of pre-mining microseisms is clustering. The dense zone of microseismic activity always formed in the area with relatively developed fractures, relatively broken rock mass, strong water abundance, and a large slurry consumption. The clustering of pre-mining microseisms could be used to reflect the zones where fractures are relatively developed and a water-conducting channel is easily formed.

One of the characteristics of pre-mining microseisms was periodicity. The energy of pre-mining microseisms was in a phased change from 'low frequency and small energy' to 'high frequency and small energy' to 'high frequency and large energy', so as to complete

the release of energy for rock failure. This change process corresponded exactly to the multiple phases of the grouting process (filling grouting, penetrative grouting, fracturing grouting). The periodicity of the pre-mining microseisms applied to the monitoring of a single branch hole grouting and regional grouting process, and the evaluation of the grouting effect.

## 4. Correlation Analysis between Characteristic of Pre-Mining Microseismic and Water Abundance of Floor Aquifer

The Shanvuqing aquifer is a direct water-filled aquifer of coal seam 2 on the 182602 working face. The water abundance of the Shanvuqing aquifer is strong and heterogeneous. It is laterally recharged by Ordovician limestone water, which might still pose a threat of water inrush to coal seam mining after grouting reinforcement. The accuracy of the exploration results of the aquifer by geophysical methods before mining on the working face remains to be discussed [24]. Therefore, it is of great significance to understand the correlation between the characteristics of pre-mining microseisms and the water abundance of the Shanvuqing aquifer.

### 4.1. Analysis of Water Abundance of the Shanvuqing Aquifer

The average total thickness of the Shanvuqing aquifer is 5.5 m, with local water abundance. According to the actual exposure of the boreholes within the 182602 working face, the average distance from the aquifer to the bottom of the working face is 58 m, the maximum water inflow from a single hole is 80 $m^3$/h, the highest water level of the aquifer is +113.5 m, and the water temperature is 36.5 °C~45.2 °C. The 182602 working face was separated into the inner and outer sections based on the distribution of the water abundance of the aquifer. The basic conditions of the Shanvuqing aquifer within the inner and outer sections of the 182602 working face are shown in Table 3.

**Table 3.** The basic conditions of the Shanvuqing aquifer within the inner and outer sections.

| Zone | Location in Relation to the Zhonglicun Syncline | Number of Boreholes | Water Inflow ($m^3$/h) | Water Pressure (MPa) | Water Temperature (°C) | Water Level (m) |
|---|---|---|---|---|---|---|
| Inner section | Away from the side | 53 | 0.1~80 | 1.3~8.1 | 36.5~44.1 | −606.5~113.5 |
| Outer section | Near the side | 121 | 0.2~80 | 1.7~8.9 | 38.2~46.5 | −568.5~102.5 |

We commonly assumed that the water inflow, water pressure, and grouting volume were the outward performance characteristics of the water abundance of the Shanvuqing aquifer [25].

(1)  The water inflow of boreholes

The fracture development degree of the aquifer is closely related to its water storage capacity, and also determines the recharge and discharge conditions of the aquifer. Since the fracture development degree of the aquifer is difficult to measure by direct means, the water inflow of the boreholes can reflect the fracture development degree and water abundance of the Shanvuqing aquifer.

According to engineering practice, a water inflow higher than 10 $m^3$/h is called high water inflow. As shown in Figure 9c, there were 12 boreholes with a water inflow not less than 10 $m^3$/h within the inner section, accounting for 25% of the total boreholes, and the maximum water inflow reached 50 $m^3$/h. There were 55 boreholes with a water inflow not less than 10 $m^3$/h in the outer section, accounting for 46.2% of the total boreholes, and the water inflow of several boreholes reached 60 $m^3$/h, with the maximum water inflow reaching 80 $m^3$/h. Therefore, the boreholes with higher water inflow in the outer section were significantly more than those in the inner section.

(2)  The water pressure of boreholes

The water pressure value of the aquifer is one of the main elements reflecting the water abundance. When the water pressure value is higher, it indicates that the water abundance of the aquifer is stronger.

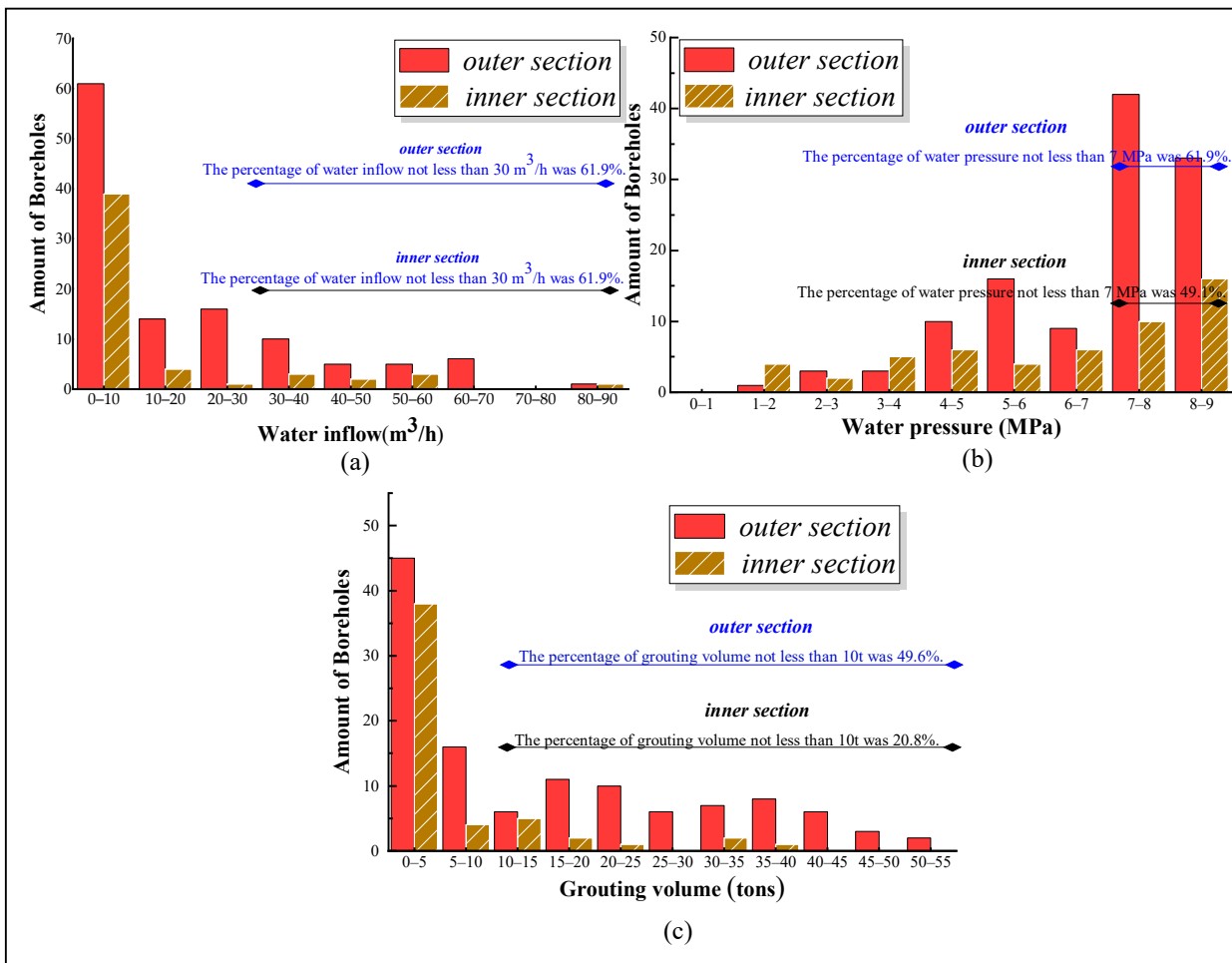

**Figure 9.** Variation in the water inflow, water pressure, and grouting volume in the inner and outer section.

According to engineering practice, a water pressure higher than 7 MPa is called high water pressure. As shown in Figure 9b, there were 25 boreholes with a water pressure not less than 7 MPa within the inner section, accounting for 49.1% of the total boreholes, and the maximum water pressure reached 8.1 MPa. There were 75 boreholes with a water pressure of not less than 7 MPa in the outer section, accounting for 61.9% of the total $n\backslash$boreholes, and the maximum water pressure reached 8.9 MPa. Therefore, the boreholes with higher water pressure in the outer section numbered significantly more than those in the inner section.

(3) The grouting volume of boreholes

The grouting volume of boreholes can reflect the degree of fracture development of the aquifer to a certain extent. When the grouting volume is greater, the fractures are developed and the connectivity is good.

According to engineering practice, a grouting volume higher than 10 tons is called a high grouting volume. As shown in Figure 9c, there were 10 boreholes with grouting volume not less than 10 tons within the inner section, accounting for 20.8% of the total boreholes, and the grouting volume reached 23.52 tons. There were 60 boreholes with a grouting volume not less than 10 tons in the outer section, accounting for 49.6% of the total boreholes, and the maximum grouting volume reached 56.7 tons. Therefore, the boreholes

with a higher grouting volume in the outer section numbered significantly more than those in the inner section.

Therefore, the boreholes with a high water inflow, high water pressure, and high grouting volume (hereinafter referred to as '3-high' boreholes) in the outer section numbered significantly more than those in the inner section. Therefore, the water abundance of the outer section was significantly higher than that of the inner section, showing a gradually decreasing trend from the east to the west of the working face, from the syncline axis to the open-cut of the working face.

### 4.2. Correlation Analysis between Microseismic Events and '3-High' Boreholes

The number of '3-high' boreholes could indicate the water abundance of the Shanvuqing aquifer in different zones. We investigated the relationship between the water abundance and microseismic events along the strike of the working face. As shown in Figure 10, the variation in the water inflow, water pressure, water temperature, and grouting volume was basically positively correlated, and the trend of microseismic events along the strike of the working face was consistent with that of the '3-high' boreholes. Microseismic events and '3-high' boreholes increased gradually along the strike of the working face. Microseismic events and '3-high' boreholes in the outer section numbered significantly more than those in the inner section, and increased significantly in the fracture zone around the Zhonglicun syncline.

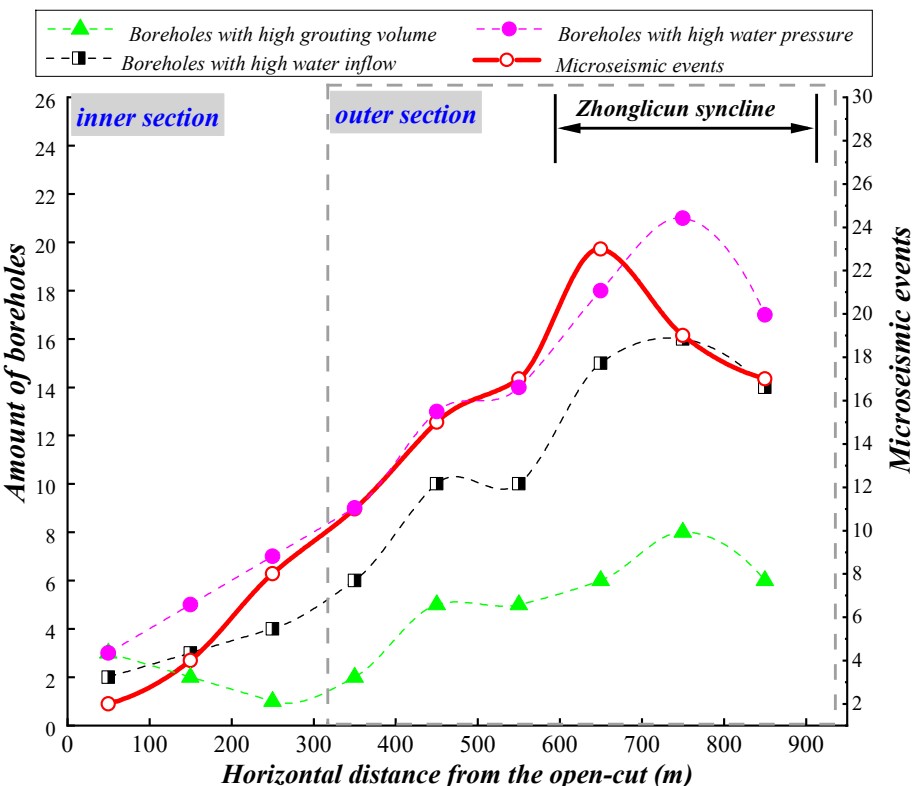

**Figure 10.** The variation trend of '3-high' boreholes and microseismic events along the strike.

The correlation coefficient in statistics can be used to reflect the closeness of the correlation between variables.

$$\gamma(X,Y) = \frac{Cov(X,Y)}{\sqrt{Var[X]Var[Y]}}$$

where $Cov(X,Y)$ is the covariance of $X$ and $Y$; $Var[X]$ is the variance of $X$; $Var[Y]$ is the variance of $Y$.

The correlation coefficients $\gamma_1$, $\gamma_2$, and $\gamma_3$ were calculated for the microseismic events and the boreholes with high water inflow, high water pressure, and high grouting volume, respectively, and the results can be obtained as $\gamma_1 = 0.86$, $\gamma_2 = 0.82$, and $\gamma_3 = 0.71$. Therefore, microseismic events were highly correlated with the distribution of '3-high' boreholes, that is, pre-mining microseisms were highly correlated with the water abundance of the Shanvuqing aquifer.

In conclusion, microseismic events within the Shanvuqing aquifer gradually decreased from the direction of the Zhonglicun syncline axis to the open-cut of the working face, which was essentially consistent with the distribution of the water abundance of the Shanvuqing aquifer. Microseismic events were more concentrated in zones with high water inflow, high water pressure, and high grouting volume, and microseismic events were significantly reduced in zones with weak water abundance. Therefore, pre-mining microseisms can identify the spatial water abundance differences of the main water-filled aquifers and characterize the water inrush risk zones.

### 4.3. Risk Assessment Method of Water Source along the Strike of the Working Face

Under the assumption that all other conditions were essentially the same, based on the elements of cumulative water inflow, average water pressure, and cumulative grouting volume of the grouting boreholes, a risk grading criteria of water source is presented as it was feasible to measure the risk level of the floor water inrush by judging the water source and its water abundance.

The risk of the water source was graded regarding the following criteria. The zone with a high water inflow, high water pressure, and high grouting volume was the dangerous zone. When only one or two of the three elements was high, the zone was suspected as being dangerous. Since pre-mining microseisms were highly correlated with the water abundance of the aquifer, the water inrush risk should be subsequently verified by the distribution of microseismic events. Furthermore, zones with low water inflow, low water pressure, and low grouting volume were the safe zones.

The information of 174 grouting boreholes was counted and cumulative water inflow, average water pressure, and cumulative grouting volume were examined along the strike, as shown in Figure 11. According to the risk grading criteria of the water source, there was no dangerous zone in the inner section, and there were two danger zones in the outer section, which were located in the fault F602-4 and Zhonglicun syncline, respectively. The zone at a distance of 0–50 m from the open-cut was expressed in the form of high cumulative water inflow, low average water pressure, and high cumulative grouting volume. In addition, there was only one element with a large value at a distance of 250–300 m, 350–400 m, 450–500 m, and 750–800 m from the open-cut, which were suspected as being dangerous zones. The rest were normal zones.

The microseismic distribution is shown in Figure 12. A total of 75.3% of the microseismic events were clustered within the dangerous zones of the outer section; 5.2% of the microseismic events were sporadically distributed in the suspected dangerous zones of the inner section, indicating that the risk of floor water inrush was at a low level; and 13.3% of the microseismic events were distributed in the suspected dangerous zones of the outer section, indicating that the risk of floor water inrush was at a high level. Therefore, when mining the outer section of the working face, exploration and water control measures should be conducted in advance.

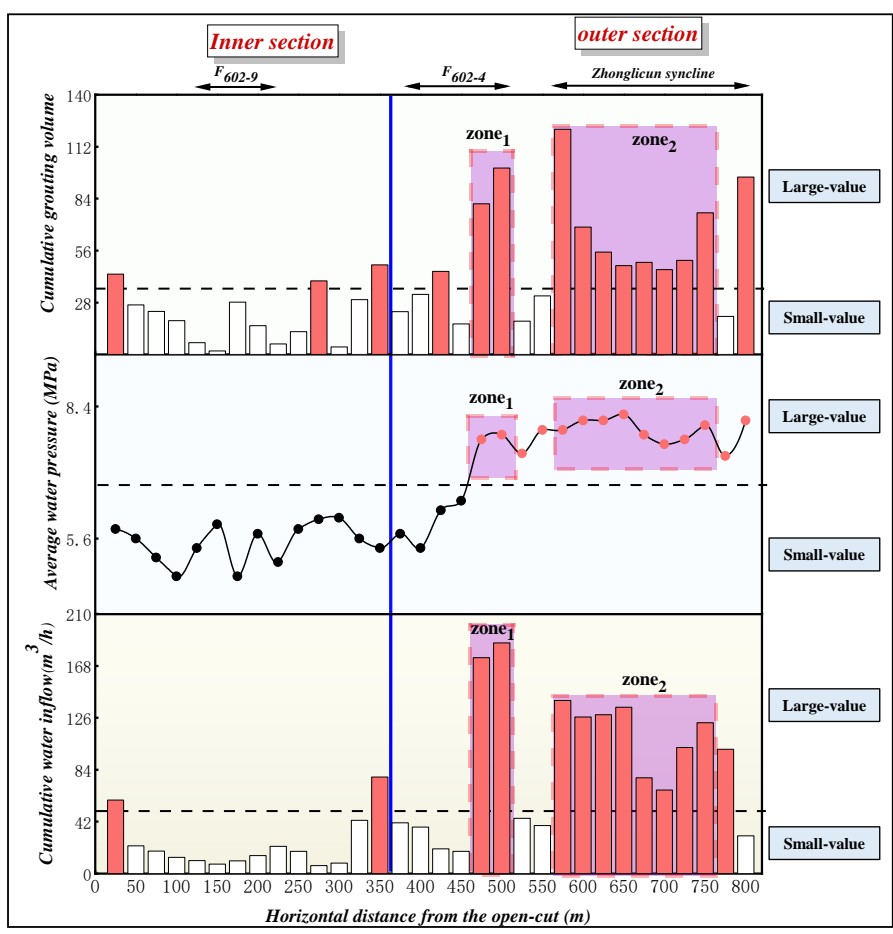

**Figure 11.** The variation trend of the three elements s along the strike of the working face.

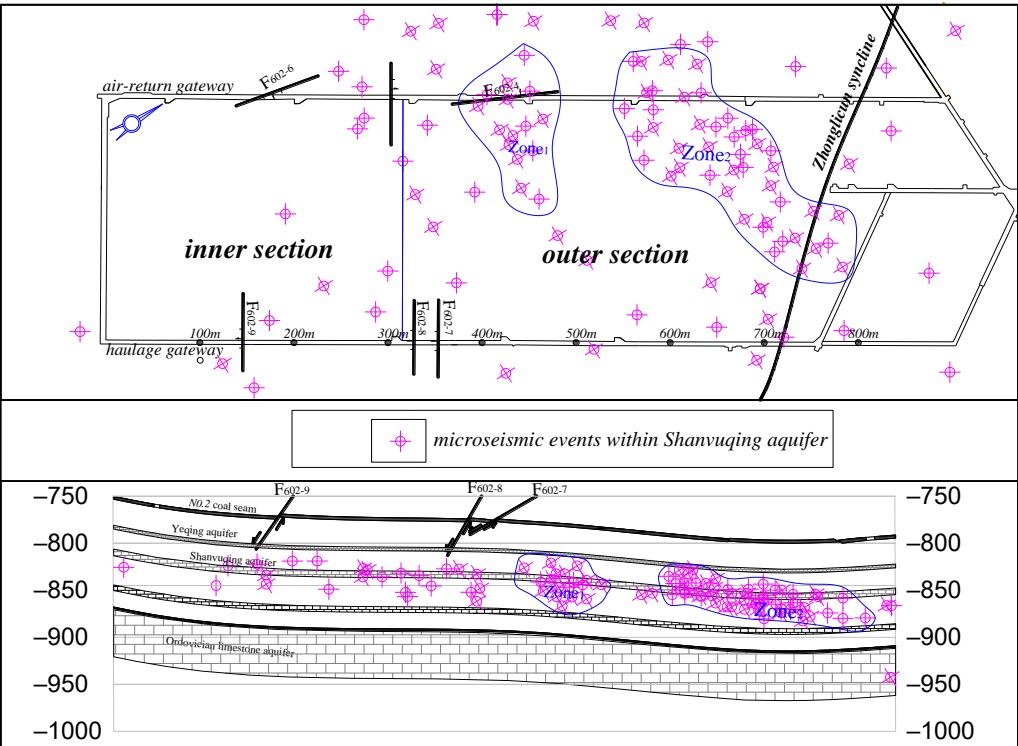

**Figure 12.** The distribution of microseismic events within Shanvuqing aquifer.

## 5. The Vulnerability Index Model of Floor Water Inrush Based on Level Analysis and Entropy Weight Method

The vulnerability index approach [26] is a new practical method for the risk evaluation of floor water inrush that takes into account numerous parameters (including their respective weights) and can reflect the nonlinear dynamic phenomenon of floor water inrush under the influence of multiple factors. We built the vulnerability index model for the risk of floor water inrush using the microseismic monitoring results and the water abundance of the aquifer as research indicators. The determination of weights in the modeling process was challenging, since it is erroneous to adopt a single objective or subjective weighting technique for the risk evaluation of floor water inrush. This might result in too much subjective arbitrariness in the evaluation results or too much concentration on the data itself. As a result, we first utilized the level analysis to determine the weights and subjectively analyzed the value of each index. Next, we applied the objective entropy weight method to estimate the relative relevance of the indicators and assigned weights, then we assigned the weights to establish the comprehensive weights, which improved the accuracy and reliability of the evaluation results [27,28]. The model building process is shown in Figure 13.

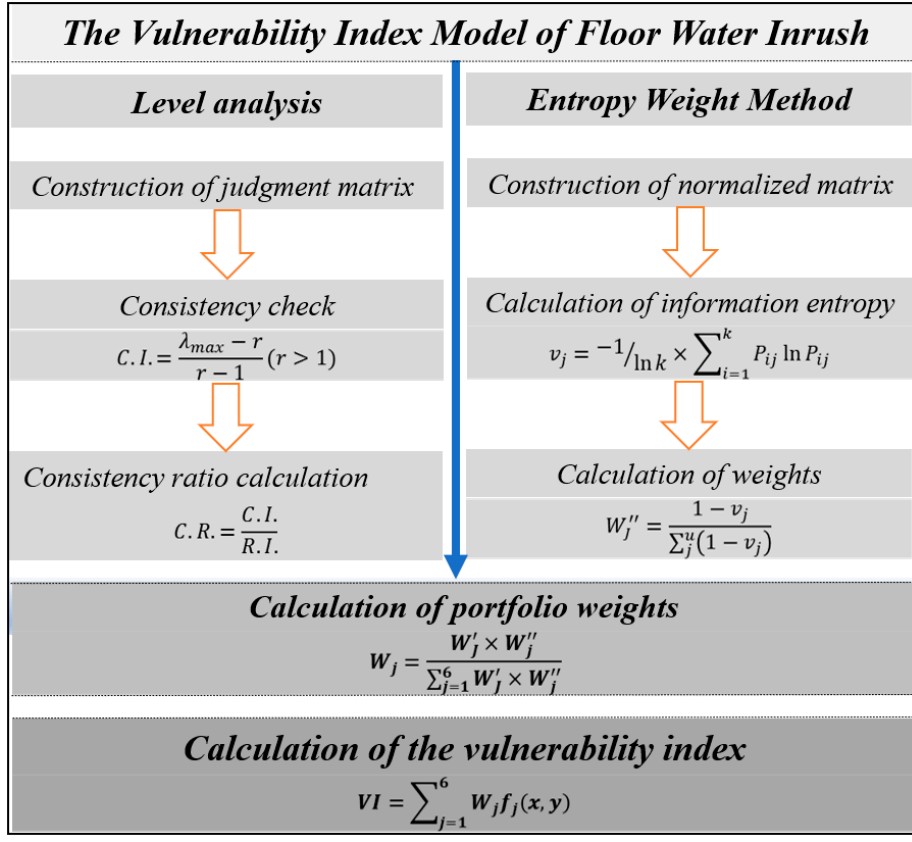

**Figure 13.** The vulnerability index model of the floor water inrush building process.

### 5.1. Analysis of the Risk of Floor Water Inrush within the Range of the 182602 Working Face

The microseismic kernel density, three elements of water abundance (water volume, water pressure, grouting volume), and two elements of tectonic (fault, syncline) explained the floor rock state and the risk of floor water inrush from diverse angles. According to the preceding paragraph, one of the characteristics of pre-mining microseismic events was clustering, the dense zone of microseismic activity was always formed in the zones with relatively developed fractures, relatively broken rock strata, and a high water abundance under the action of artificial disturbance. Therefore, the higher the value of the microseismic kernel density, the higher the fragility of the rock strata and the higher the risk of floor water inrush. In addition, zones with high water abundance had a high risk of floor water inrush.

In the tectonic zones, fractures were well-developed, which were easy to communicate with the Ordovician limestone aquifer below, and the risk of water inrush was high.

We examined the risk of floor water inrush within the 182602 working face. First, the above six elements were used as research indicators, and the weights $W'$ and $W''$ were determined from the subjective and objective perspectives by using level analysis and the entropy weight method, respectively, and the comprehensive weight $W$ was determined. After that, the vulnerability index model was used to evaluate the risk of floor water inrush. Finally, the risk zoning evaluation chart was drawn with the help of ArcGIS technology. The comprehensive weights were calculated as shown in Table 4.

**Table 4.** The comprehensive weights of the six research indicators.

| Research Indicators | $W'$ | $W''$ | $W' \times W''$ | $W$ |
| --- | --- | --- | --- | --- |
| Microseismic kernel density | 0.202 | 0.213 | 0.0345 | 0.1866 |
| Water inflow (m$^3$/h) | 0.247 | 0.268 | 0.0637 | 0.3442 |
| Water pressure (MPa) | 0.213 | 0.175 | 0.0373 | 0.2014 |
| Grouting volume (tons) | 0.084 | 0.063 | 0.0097 | 0.0522 |
| Fault (m) | 0.072 | 0.057 | 0.0073 | 0.0397 |
| Syncline (m) | 0.182 | 0.224 | 0.0326 | 0.1759 |

The product of the comprehensive weights of the research indicators and the quantified normalized value were calculated, that is, the vulnerability index model of the floor water inrush was established. The *VI* values were calculated and counted, and the normalized thematic charts of the research indicators were drawn by using information processing technology, as shown in Figure 14.

$$VI = 0.1866 f_1(x,z) + 0.3442 f_2(x,z) + 0.2014 f_3(x,z) + 0.0522 f_4(x,z) + 0.0397 f_5(x,z) + 0.1759 f_6(x,z) \quad (1)$$

where *VI* is the risk degree indicator; $f_j(x,z)$ is the quantified normalized value of the *j* influencing elements.

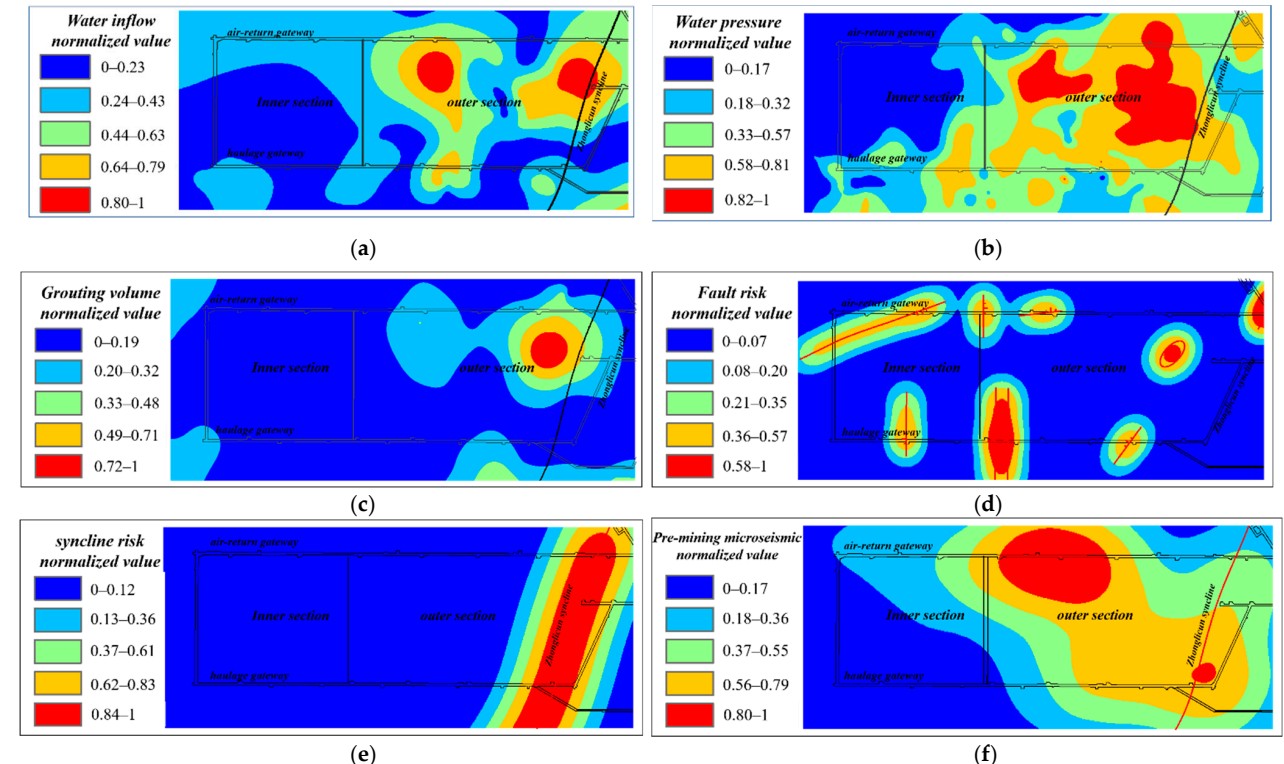

**Figure 14.** Normalized thematic charts of the research indicators. (**a**) Water inflow, (**b**) Water pressure, (**c**) Grouting volume, (**d**) Fault, (**e**) Syncline, (**f**) Pre-mining microseisms.

The consequence of the combined influence of several study indicators was the floor water inrush, which is a complex process governed by multiple elements. As a result, the thematic charts of each research index were superimposed to create the final comprehensive evaluation chart of floor water inrush, and natural breaks (Jenks) were used to categorize the comprehensive evaluation of floor water inrush into five levels: level I–level V, with level I being the most dangerous zone and level V being the relatively safe zone, as shown in Figure 15.

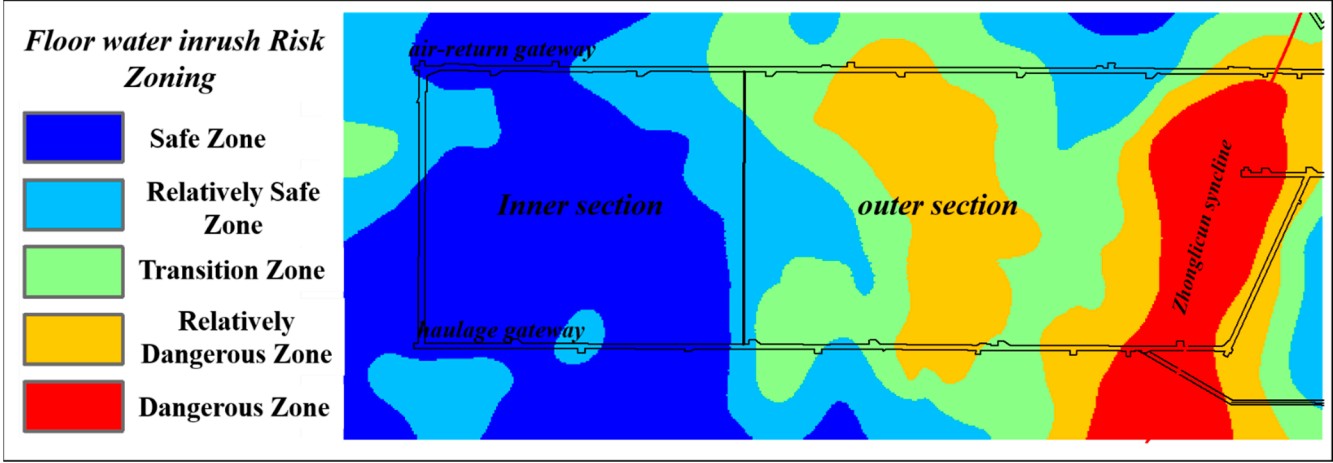

**Figure 15.** The comprehensive risk evaluation of the floor water inrush within the 182602 working face.

It can be seen from the figure that the water inrush dangerous zone and relatively dangerous zone were mainly distributed in the outer section near the Zhonglicun syncline side. Compared with the calculation results of the water inrush coefficient, dangerous and relatively dangerous zones were also the zones with the maximum water inrush coefficient (aa water inrush coefficient greater than 0.06). The dangerous and relatively dangerous zones were also zones with high water abundance. At the same time, the dangerous and relatively dangerous zones were also zones with the most concentrated pre-mining microseisms. According to the analysis, the hydrogeological characteristic in the two flanks of the Zhonglicun syncline was complex, making it the most likely zone for floor water inrush. Therefore, when the working face was mined to this zone, we should pay close attention to the changing trend of water inflow in the goaf, make accurate judgments in combination with the actual situation of the site, and specify reasonable water inrush countermeasures.

### 5.2. Geophysical Exploration of the Risk of Floor Water Inrush

To make the vulnerability index model of floor water inrush based on pre-mining microseisms more trustworthy and to investigate the hydrogeological conditions in the working face and the surrounding area, we employed the network parallel electrical method to conduct the physical analysis of the floor rock layer [29]. The method was used to obtain the apparent resistivity of the rock mass according to the conductivity difference between the rocks and ore bodies. Low apparent resistivity represents the high degree of fracture and water abundance of the rock layer in the exploration zones.

We used the YBD11 mining network parallel electrical instrument to place electrodes and an electrical survey line in the haulage gateway, air-return gateway, and open-cut in turn, with a total of 13 exploration points and 64 electrodes per exploration point. According to previous exploration experience, the apparent resistivity of the abnormal zones of the Wutongzhuang mine was generally below 20 Ω·m. The results of the geophysical exploration are shown in Figure 16. In the vertical direction, the abnormal zones were mainly developed in the Shanvuqing layer. In the horizontal direction, there were two

abnormal zones DF1 and DF2 in the inner section, and three abnormal zones DF3, DF4, and DF5 in the outer section of the working face. However, the results of the borehole exploration and verification showed that DF1 and DF2 in the inner section were weakly water abundance zones and DF3, DF4, and DF5 in the outer section were high water abundance zones. Compared with the vulnerability index model, the spatial distribution of the abnormal zones DF3, DF4, and DF5 were found to be essentially the same as that of the water inrush dangerous zone and relatively dangerous zone. In conclusion, the microseismic-based water risk assessment method is accurate and dependable, and it may be used to guide and inform water management in coal mines.

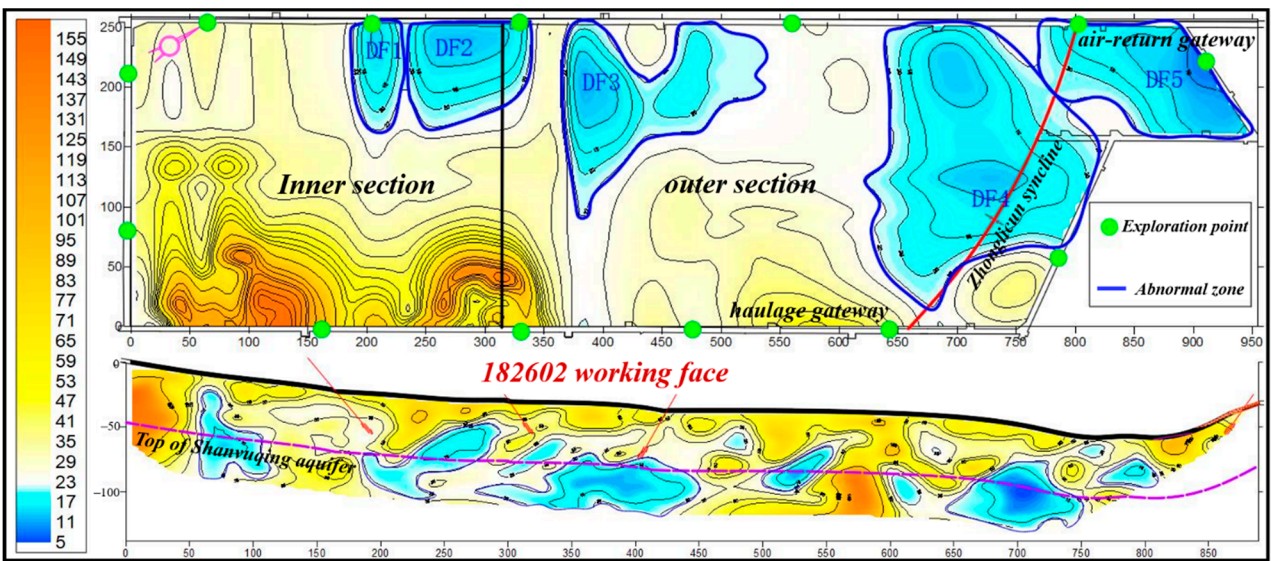

**Figure 16.** The geophysical exploration of the risk of floor water inrush.

## 6. Discussion

Pre-mining microseisms will be triggered in the fracture development zones under artificial disturbance such as grouting works. The high-pressure slurry caused local stress concentration in the fractures and cavities in the formation, so the cracks and cavities split and expanded. The elastic strain energy released during this process led to the occurrence of microseismic events. As the grouting work progressed, the fracture network was filled and the formation was strengthened. The water-conducting channel of groundwater was blocked and the groundwater flow field was rebalanced, greatly reducing the nuclear density of microseismic events. Based on this formation mechanism of the pre-mining microseisms, a vulnerability index model of floor water inrush was built using level analysis and the entropy weight method. The 182602 working face has had second floor grouting reinforcement carried out in the predicted dangerous zones. Twenty-six boreholes have been constructed, and the final hole depth exceeded the Shanvuqing aquifer. Through the targeted grouting treatment measures, the efficient and safe mining of the working face was finally realized.

Unfortunately, the microseismic monitoring system is often installed just as the working face is about to be mined, which makes it very difficult to monitor the pre-mining microseismic events during the grouting period. Limited data were used, which reduces the accuracy and reliability of this study. In subsequent research, we will focus on new study areas such as the Jiaozuo mining area and Xingtai mining area, and further deepen the law and mechanism of pre-mining microseisms.

## 7. Conclusions

In this paper, taking the 182602 working face of the Wutongzhuang coal mine as the research background, combined with the plane, spatial, and energy characteristics of microseismic events before mining, the concept of pre-mining microseisms was proposed.

On this basis, the correlation between microseismic events and the water abundance of the aquifer was considered, and an evaluation method with regard to floor water inrush was constructed based on "three elements" of the aquifer and pre-mining microseisms. The main conclusions are as follows.

(1) Under the artificial disturbances including slurry diffusion and neighboring mining, there were relatively dense microseismic events in the zones where water-conduct channels are prone to occur. This type of microseismic event can be called pre-mining microseisms, which is characterized as being "sporadic, cluster and phased".

(2) It was commonly assumed that the obvious performance characteristics were water inflow, water pressure, and grouting volume in the water abundance of the Shanvuqing aquifer. It was verified that the water abundance of the outer section was significantly higher than that of the inner section. The correlation between the microseismic events and three high drilling holes was noticeable with values of 0.86, 0.82, and 0.71, respectively. On this basis, a risk assessment method of water source along the strike of the working face was proposed. Two dangerous zones were determined.

(3) According to the generated vulnerability index model, the dangerous zone and relatively dangerous zone were determined comprehensively in the outer section of the working face, which were basically consistent with the observation result of the DC method. As such, the zones were at risk of forming water-conducting channels.

**Author Contributions:** L.H. and Q.G., methodology; L.H. and Y.X., writing—review and editing; L.H. and S.L., formal analysis; Y.L. and L.Z., data curation and validation; W.M., original draft preparation. All authors have read and agreed to the published version of the manuscript.

**Funding:** This research was funded by the Fundamental Research Funds for the Central University of China (2022YQNY04) and Key Project of Research and Development Plan of Hebei, China (22375422D).

**Institutional Review Board Statement:** Not applicable.

**Informed Consent Statement:** Informed consent was obtained from all subjects involved in the study.

**Data Availability Statement:** The data that support the findings of this study are available from the corresponding author, upon reasonable request.

**Acknowledgments:** The authors gratefully acknowledge the monitoring equipment of Hebei Coal Research Institute Co., Ltd. The authors gratefully acknowledge the help and comments of the reviewers and editors.

**Conflicts of Interest:** The authors declare no conflict of interest.

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
