# Peer review of "Research on the Development Law of Pre-Mining Microseisms and Risk Assessment of Floor Water Inrush: A Case Study of the Wutongzhuang Coal Mine in China"

_sustainability, doi:10.3390/su14159774_

Round 1

Reviewer 1 Report

Overall good work and interesting analyses. I have only a few comments and suggestions.

1.  The downloaded manuscript only had one of the figures included in Figure 14. Figure 14a was included, but the others are not present.

2. Please check that all variables C.I.  W, and others have been defined for the reader in the text or in the notes for figures and tables.

 3. Is depressurizing the aquifer, that is pumping to reduce the hydraulic head, a potential treatment for this mine?  Should that be discussed?

4.  I like the figures and graphics. Thye convey a large amount of useful information and make the manuscript easier to read.

Reviewer 2 Report

This article is read with interest and logically sustained. The article presents a detailed review of the studies carried out to study the relationship between the microseismic background and hydrogeological conditions. However, a number of questions arise as you read it. Unfortunately, there is no information about filtration parameters, characteristics and regime of the three aquifers, located in the zone of the coal deposit. It is recommended to submit an overview hydrogeological map with a section within which the development of a coal deposit is carried out. The geological column at Figure 3 is shown without stratification (geological indexs) and reference to the location. The relationship between the vertical scale of the block diagram and the reduced thicknesses of the layers is not entirely clear, as the average depth of the mine is 850 m. The text does not describe the F602-3 - F602-9 faults, although these zones manifest themselves differently in geophysical fields at Figures 12 and 16. Distribution of microseismic events at Figures 5 and 6  interconnected with the time and direction of well cementing, which is especially clearly seen in the example of well 7 in September. In this regard, it is probably not entirely correct to use the obtained data to identify a relatively dangerous zones in the center of the working at Figure 15. Line of cross-section  does not show at Figure 12. Figures 14 b-f are missing. In spite of remarks, the presented results are of practical importance and the article is recommended for publication with additions and corrections.

Reviewer 3 Report

The paper assesses the distribution pattern of the micro-seismic events happening in the floor. The analysis helped with determining the hydrogeological character of the formation and estimating the location of the events. The paper can be improved, considering my comments (see attached). 

Round 2

Reviewer 3 Report

The paper has been improved. The authors may proofread the paper for further improvement.